# Evaluating Targeted Intervention on Coal Miners’ Unsafe Behavior

**DOI:** 10.3390/ijerph16030422

**Published:** 2019-02-01

**Authors:** Ruipeng Tong, Yanwei Zhang, Yunyun Yang, Qingli Jia, Xiaofei Ma, Guohua Shao

**Affiliations:** School of emergency management and safety engineering, China University of Mining and Technology (Beijing), Beijing 100083, China; 13121957417@163.com (Y.Z.); yangyunyun0225@126.com (Y.Y.); jiaqingli123456@126.com (Q.J.); 18811739823@163.com (X.M.); 18810700823@163.com (G.S.)

**Keywords:** unsafe behavior, targeted intervention, coal miners, safety management

## Abstract

Miners’ unsafe behavior is the main cause of roof accidents in coal mines, and behavior intervention plays a significant role in reducing the occurrence of miners’ unsafe behavior. However, traditional behavior intervention methods lack pertinence. In order to improve the intervention effect and reduce the occurrence of coal mine roof accidents more effectively, this study proposed a targeted intervention method for unsafe behavior. The process of targeted intervention node locating was constructed, and based on the analysis of 331 coal mine roof accidents in China, three kinds of targeted intervention nodes were located. The effectiveness of targeted intervention nodes was evaluated by using structural equation model (SEM) through randomly distributing questionnaires to miners of Pingdingshan coal. The results show that, in preventing roof accidents of coal mines, the targeted intervention nodes have a significant positive impact on the intervention effect. The method can also be applied to the safety management of other industries by adjusting the node location and evaluation process.

## 1. Introduction

At present, roof accidents are still frequent in coal mine production. According to a previous analysis of roof accidents in coal mines, engineering technical means cannot completely control the occurrence of roof accidents. However, research on the influence of human behavior is not clear [1,2,3]. Currently, the research on roof accidents is mainly divided into two categories: management and technology. The former does not point out concrete operation mistakes or management mistakes, and is a barrier to measuring formulation pertinence. The latter is combined with specific coal mining faces or roadways, so the mechanism of roof deformation and failure instability has been studied, and relevant engineering measures have been worked out [4]. However, despite the continuous improvement of the technical level of roof support in China, roof accidents still occur from time to time. This shows that engineering technical means are not the best measures to solve roof accidents. With continuous improvement and a deepening of people’s understanding and the research on behavior safety, researchers have gradually discovered that unsafe behavior is a more significant cause of accidents [5]. Similarly, the vast majority of coal mine casualties are caused by unsafe behavior by miners [6]. Unsafe behavior refers to that that may cause casualties, property damage, and environmental damage in violation of rules of operation and safety regulations. In order to effectively prevent roof accidents, human factors must be taken into account. Combined with BBS (Behavior Based Safety) theory, explicit behavioral intervention measures should be put forward [7]. Reducing the incidence of unsafe behavior among workers through behavior intervention is conducive to reducing the risk of enterprise safety, thus enhancing the effectiveness of the enterprise safety management system [8].

A miner’s behavior is based on a complex decision-making process, and behavior safety research is mainly based on objective theory and operational condition theory. The emphasis is to identify key unsafe behavior and correct unsafe behavior through intervention. Currently, the research on unsafe behavior intervention mainly explores the intervention countermeasures of unsafe behavior by combining organizational factors with individual factors [9]. Using planning behavior theory, accident cause theory, the structural equation model, and other methods, research has been done on coal mines, buildings, aviation, and other fields, respectively, with respect to safety training, safety culture, performance feedback, material incentives, and other aspects of appropriate intervention measures to reduce the occurrence of unsafe behavior [10,11,12]. Namian et al. found that the prolonged use of ineffective safety training methods seriously affected the safety in a building. They collected safety training data at the project level to measure workers’ risk identification ability and safety risk perception level, and the results provided references for improving safety training work. It shows that traditional safety training methods, to enhance their effectiveness, need to be combined with modern information technology [13]. Kouabenan’s research on two French nuclear power plants proves that the safety climate has a certain substantial impact on the promotion of safety management, but it was also found that the encouragement of the direct supervisor in the enterprise is more influential than the view of senior management on safety. This also indicates that more in-depth research on human behavior characteristics is needed in enterprise security management [14]. Warszawska et al. found that a weak safety culture is the main cause of many catastrophic events and that, in order to avoid this situation, enterprise safety culture must be strengthened [15]. These traditional methods of unsafe behavior intervention are mostly based on the observations and records of workers’ behaviors, the assessment of behavioral risks to overall conduct, and extensive behavior intervention of workers. In general, traditional methods are only used for the intervention of unsafe behavior itself and lack the in-depth analysis of its internal characteristics and the in-depth study of the root factors leading to unsafe behavior [16]. Therefore, this intervention method lacks pertinence. People’s unsafe behavior cannot be fundamentally changed, and the root factors of unsafe behavior will reappear after the intervention [17].

With the continuous expansion of the data volume of security information, analyzing and mining the hidden value of the data have become important in behavior security research [18]. The purpose of this study is to propose a targeted intervention method for unsafe behavior on the basis of behavior safety management. This method pays more attention to the inherent characteristics of workers’ unsafe behaviors on the basis of traditional intervention methods through data mining to fully master the risk level, position, behavior individual, behavior trace, behavioral property, time, and type of unsafe action dimension information for worker’s unsafe behavior. Statistical analyses and the association rule mining method are applied to analyze various dimensions of information regarding distribution and the inner link between them. According to the internal information of unsafe behaviors, intervention nodes are located. The ultimate goal is to improve the effect of behavioral intervention by formulating corresponding intervention measures for each targeted intervention node in actual safety management work. Through data analysis of coal mine roof accident cases, three types of targeted intervention nodes are identified: the key types of behavior-targeted intervention nodes, the single-dimensional characteristic-targeted intervention nodes, and the multi-dimensional characteristic-targeted intervention nodes. The structural equation model (SEM) was used to evaluate each targeted intervention node to prove the effectiveness and practicability of the method.

## 2. The Targeted Intervention Nodes of Miners’ Unsafe Behavior

The significance of “targeted” has different meanings in different industries, and in medicine, marketing, and economic development, it mainly refers to targeted drugs, targeted marketing, and targeted poverty alleviation. “Targeted” used in the field of behavior safety management refers to the realization of precise positioning, precision intervention, and the accurate management of unsafe behavior of miners based on existing research. The most critical step to achieve targeted intervention is to locate the targeted intervention node through data mining, statistical analysis, and other methods to identify the key behavior of each specialty and the characteristics of the single dimensional, multi-dimensional, and other deep-seated information [19]. Finally, the corresponding intervention measures are made for each intervention node, so as to correct the unsafe behavior of miners and prevent accidents.

The work of coal mine safety management is difficult, and the unsafe behavior of miners is complex and diverse [20]. Traditional behavioral interventions often use methods such as material reward, goal setting and performance feedback, management intervention, and so on. The accuracy of “netting” and “one-size-fits-all” intervention methods is low, and they not only cannot fully capture the unsafe behavior of the site but will also certainly lead to a miner’s aversion and a waste of money. At present, with the construction of digital mines, coal mines have realized the overall perception of underground personnel, equipment and facilities, the environment, and other objects, and the accident reports and various rules and regulations have also gradually improved. This can provide a large amount of data support for the location of target nodes. Based on the above analysis, the process of targeted intervention nodes for unsafe behavior of miners was constructed, as shown in Figure 1.

### 2.1. Data Sources

The hidden danger of job sites and violation images can be used as realistic data, recording the occurrence time of the accident, the behavior process of the individual, and the situation of all things, which can clearly and truly reflect the unsafe behavior and the safety risk points of miners [21]. The accident report records the process, the cause, the corrective action, and the result of the accident. Many regulations and standards provide guidance for the standardized description of unsafe behavior, which can be used as abstract data for data analysis. 

### 2.2. Data Boundary

Seven dimensions are used to describe the unsafe behavior comprehensively, and the information of unsafe behavior in the form of text and pictures is transformed into structured data by coding. Risk level (RL) indicates that the severity of unsafe behavior is divided into three levels: high-risk, medium-risk, and low-risk; position (P) describes the location where unsafe behavior occurs; behavior individual (BI) expresses the information of the person who issued the unsafe behavior, which includes age, length of service, educational background, and so on; behavior trace (BT) shows whether the unsafe behavior can be traced after its occurrence and whether the behavior can be divided into traced unsafe behavior and non-traced unsafe behavior; behavior property (BP) is divided into four parts: violation of command, violation of operation, violation of action, and non-violation unsafe action; time (T) records when unsafe behavior occurs; unsafe action (UA) describes the specific unsafe behavior that may lead to accidents, casualties, and environmental disruption.

### 2.3. Data Analysis

Through the single-dimensional analysis of unsafe behavior data, the distribution characteristics of risk level, position, behavior individual, behavior trace, behavior property, time, unsafe action, and specialty have been explored. Through the interaction analysis of each dimension, the interaction rules of unsafe behavior between different dimensions are explored. Data analysis provides sufficient evidence for locating the targeted intervention nodes.

### 2.4. Locating Nodes

Firstly, the key behavior of each position should be determined. The targeted intervention nodes oriented to the key behavior of different types of work will be located. Secondly, the distribution differences of miners’ unsafe behavior in one particular dimension and different categories will be studied, and the distribution characteristics in the specific dimension are obtained, so as to locate the targeted intervention node oriented to the single-dimensional feature of unsafe behavior. The occurrence of miners’ unsafe behavior has its inherent complexity, which is influenced by the natural environment, geological conditions, construction technology, personnel characteristics, management level, and so on [22]. The interaction between variables in theory will be combined with the requirement of practical security management. The interaction between different dimensions of unsafe behavior is exploring, and the potential characteristics and deep regularity of unsafe behavior should be found. Finally, targeted intervention nodes oriented to multi-dimensional association will be located.

## 3. Research Methods

### 3.1. Unsafe Behavior Data Analysis Based on Roof Accidents 

The number of roof accidents is very large, it is very difficult to integrate the detailed case data of each accident. In order to ensure the scientific nature of the study, the selection of cases follows two principles: accident integrity and case authority. The main sources of accident cases are Internet, accident analysis report, coal mine typical accident compilation, and so on. The application of data analysis methods has the following advantages: the large amount of data, rather than the selection of random samples, greatly reduces the impact of random events on the overall conclusion; the intrinsic characteristics associated with behavior, not just superficial causality, can be studied; the processing speed is fast, and valuable information can be quickly obtained in a short period of time [23].

Unsafe behavior refers to the behavior of the person who has caused the accident or may cause the accident. There are many causes of unsafe behaviors, including individual factors, psychological factors, organizational factors, environmental factors, and so on. Different researchers have different views on the classification of unsafe behaviors. Unsafe behaviors in a coal mine mainly refer to the “three disobeying” of coal mine safety production [24]. “Three disobeying” is the general term of disobeying command, disobeying operation, and disobeying labor discipline. The elimination of “three disobeying” has always been an important issue in the safety management of all industries, especially coal mining enterprises and other high-risk industries. In order to fundamentally explore the objective rule of unsafe behaviors and the complex relationship between its internal factors, many unsafe behaviors need to be objectively analyzed. Previous coal mine accidents can provide enough data for the analysis of miners’ unsafe behaviors, but the unsafe behaviors that lead to different types of accidents are different in nature. To eliminate the heterogeneity of a large number of unsafe behavior data, we selected unsafe behavior leading to roof accidents in coal mines as the research object [25].

A total of 331 roof accidents were collected from 1983 to 2014 in China, including 8 major accidents, 159 major accidents, and 164 general accidents. All unsafe behaviors of miners causing roof accidents are collected from the accident report. For example, a roof accident occurred in a mine in Pingdingshan city, Henan province, which directly resulted in a death, several serious injuries, and a total economic loss of about RMB 585,000. According to the time of this accident, four unsafe behaviors could be analyzed successively, expressed as follows: “the top plate was out of the slag and was not withdrawn in time,” related to the coal miner; “there are no timely measures to prevent potential safety hazards,” related to the field commanders; “there are no perfect operation procedures or safety technical measures established,” related to the middle management staff; and “the arrangement of roadway is in a steep-inclined coal seam,” related to the senior management staff.

According to the data boundary divided by seven dimensions, 1215 data points were obtained. The data of miners’ unsafe behavior was transformed from unstructured text records into structured data. A preliminary single-dimensional statistical analysis of the data points was performed, as shown in Table 1.

According to the description of seven dimensions of miners’ unsafe behaviors, interactive analyses can be performed on any two or more of them. The interaction analysis between different variables can map out different practical meanings, the purpose of which is to explore the deep regularity of the unsafe behavior data of miners and to provide the basis for determining the targeted intervention nodes. Through the Apriori algorithm [26], unsafe action is the consequent of the study, and other dimensions are antecedents. The relationship between information and unsafe action in each dimension is discussed (the association between specific unsafe action and time dimension is not obvious enough and is not considered for the time being), as shown in Table 2.

Support is the ratio of the number of consequents and antecedents to the total data set in the database, and the probability of their occurrence is determined. Confidence is the ratio of the support for the occurrence of a consequent and an antecedent to the support for the consequent. It is used to denote the probability of an antecedent derived from an association rule under the condition of a consequent. Lift is the ratio of the confidence of “antecedent → consequent” to the support of the consequent. It reflects the size of the consequent influenced by the antecedent [27]. Minimum support and minimum confidence were 8% and 30%, respectively, and this information was used to obtain effective strong association rules. The results of the analysis are shown in Figure 2.

### 3.2. The Miners’ Unsafe Behavior Targeted Intervention Node Establishment

The key work behaviors are determined by the frequency, work position, work time, risk level, etc. caused by the “three disobeying” in different types of work. Following the principles of reliability, measurability, controllability, observability, etc., according to the comprehensive distribution characteristics of each dimension of “three disobeying” for various types of work, several “three disobeying” behaviors are selected as the key behaviors of various types of work. These key behaviors are targeted at the first type of intervention nodes for unsafe behavior intervention. In the process of practical safety management, it is expected to improve the effect of safety management by focusing on this kind of key work behavior. It is expected that behavior correction can be achieved by interfering with these key work behaviors, and the incidence of unsafe behavior can be greatly reduced [28]. 

A single-dimensional feature-targeted intervention node is based on the distribution characteristics and proportion of each dimension to guide the intensity of intervention and the allocation of management resources in the process of security management. In terms of time, for example, more unsafe behavior happened in January, March, and August. Safety management can target these months to strengthen the observation and intervention of unsafe behavior and can directly and effectively reduce the number of unsafe behaviors. Based on the association analysis of each dimension of unsafe behavior, the multi-dimensional characteristic-targeted intervention node was located according to their deep connection and rule, which can provide guidance and direction for practical safety management. A list of three types of intervention nodes based on miner’s unsafe behavior-targeted intervention was finally formed through data and theoretical analysis for use in this study, as shown in Table 3.

### 3.3. The Targeted Intervention Node Evaluation 

The purpose of the evaluation of targeted intervention nodes is to verify the effectiveness of each intervention node and the application value of targeted intervention nodes in the intervention of unsafe behavior of miners. In this study, the structural equation model (SEM) was constructed to evaluate the targeted intervention nodes [29]. As a general framework of statistical analysis, structural equation model (SEM) is widely used in data analysis [30,31]. 

#### 3.3.1. Variable Division and Formulate Hypothesis 

In order to evaluate the effectiveness of three unsafe behavior-targeted intervention nodes, another latent variable “intervention effect” was introduced. Unsafe behavior incidence (I1), safety management efficiency (I2), and safety climate improvement degree (I3) were selected as the observed variables of latent variable “intervention effect” [32,33,34]. K1–K5, S1–S6, and M1–M7 in the list of the targeted intervention nodes were the observed variables of the “key work behavior node,” the “single-dimensional feature node,” and the “multi-dimensional feature node,” respectively. 

In the process of coal mine safety management, in order to effectively reduce the incidence of roof accidents and improve the targeting of workers’ behavior intervention, this paper analyzes and summarizes three kinds of targeted intervention nodes: key work behavior, single-dimensional features, and multi-dimensional features. The practical significance of the three factors and their impact on the effect of intervention were analyzed, and the following hypotheses were formulated:H1. Key work behavior positively influences the intervention effect.H2. Single-dimensional features have a direct influence on the intervention effect.H3. Multi-dimensional features positively influence the intervention effect.

These three types of nodes also affect each other, and the targeted intervention on a single class of nodes will also have an impact on other types of intervention nodes. For example, a single-dimensional feature intervention node may result in a change in the incidence of key work behavior, so the following assumptions are formulated:H4. Single-dimensional features positively influence key work behavior.H5. Multi-dimensional features have a direct influence on key work behavior.H6. Multi-dimensional features have a direct influence on single-dimensional features.

#### 3.3.2. Construct Model and Questionnaire Design

Based on the following hypotheses, an initial model of miners’ targeted unsafe behavior intervention node evaluation was established, as shown in Figure 3.

According to the latent variable and observation variable, the questionnaire was designed, and the experience of other scholars was consulted and verified by experts. The questionnaire’s results were compiled with the five-point Likert scale (1–5 represents total disagreement, basic disagreement, partial agreement, basic agreement, and complete agreement, respectively) [35]. This assessment method produces better data distribution. The questionnaire consists of three parts: (1) general information and background information about the interviewees; (2) questions about the role of various targeted intervention nodes in the implementation of behavior interventions; (3) questions about the intervention effect on unsafe behavior through targeted intervention nodes.

A random sampling method was used to survey the employees in a coal mine of the Pingdingshan Coal Industry Group [36]. This coal mine is located in the northwest of Pingdingshan District, Henan province. It was founded in 1956 and put into production on 31 December 1958. It included 4468 registered workers, 376 management personnel, 281 professional and technical personnel, 12 senior titles, and 116 intermediate titles. The questionnaire survey was conducted in this coal mine where several roof accidents resulting in huge casualties and property losses had occurred. A total of 260 questionnaires were sent out, and 248 responses were received. The recovery rate was 95.38%. Among the collected questionnaires, 237 were valid, and the effective rate was 95.56%. Among the interviewees, there were 42 people with a bachelor’s degree or above, accounting for 16.15%, and 185 coal miners, accounting for 71.15%; the number of safety administrators at all levels was 38, accounting for 14.62%, and 94, accounting for 36.15%, had worked for more than 15 years.

In order to ensure the applicability and validity of the data in the questionnaire, reliability analysis and validity analysis of the questionnaire data were carried out [37]. In this study, Cronbach’s α was used to measure the reliability of setting latent variables [38]. Generally, the higher this coefficient, the higher the reliability. In the exploratory study, results of the questionnaire with a reliability up to 0.7 were acceptable. The data from the questionnaire were input into SPSS 17.0 software (International Business Machines Corporation, Armonk, NY, USA), and the calculated Cronbach’s α coefficient showed that the questionnaire data had good reliability, as shown in Table 4.

Validity analysis tests the degree to which the questionnaire reflects the objective reality. Analysis of questionnaire validity was performed via KMO and the Bartlett sphericity test [39]. Under the standard condition, when the KMO is greater than 0.7 and the concomitant probability of the Bartlett sphericity test is less than significance level 0.001, the questionnaire has high validity. The calculated result was KMO = 0.837 > 0.7, and the significance probability of the statistical value of Bartlett sphericity test was less than 0.001, indicating that the index data were normally distributed. This shows that the questionnaire has high validity. According to the above analysis, the questionnaire has good reliability and validity, and the data obtained are suitable for factor analysis.

## 4. Result and Discussion

Based on the setting latent variables and observed variables (K1–K6, S1–S7, and M1–M6), r_1_~r_4_ was set as the residual variable of the corresponding latent variable, and e_1_~e_23_ was the residual variable of the corresponding observed variable. The model was tested by IBM SPSS AMOS21.0 software (International Business Machines Corporation, Armonk, NY, USA). First, the evaluation data of the structural equation model were calculated, and the model was modified by adjusting the model to meet all kinds of fitting indexes. This paper mainly used the absolute fit index, the comparative fit index, and the parsimonious fit index to judge the fitness of the model [40]. After several revisions of the model and calculation of the fitting data, the final fitting results were obtained. The model fitting reference criteria and the final model fitting data are shown in Table 5.

The result of the model modification shows that the fitting data of the model meets the requirements of the fitting criteria, which shows that the model has good fit [41]. Based on the above analysis, the result of the final model is shown in Figure 4.

According to the final output results of the evaluation model, the path coefficients of H1–H6 were 0.766, 0.81, 0.77, 0.67, 0.48, and 0.59, respectively, which proved that H1–H6 were significant influences. Based on the path coefficients of H1–H3, the order of the effects of the three target intervention nodes was as follows: single-dimensional feature nodes > multi-dimensional feature nodes > key work behavior nodes.

The results of H1 showed that the positive influence of key work behavior on the prediction of intervention effect was supported, and the accuracy and practicability of targeted intervention nodes K1–K5 were verified. The results of Judi et al. show that the precise positioning of specific unsafe behaviors and the positive strengthening of specific safety practices can effectively reduce occupational accidents [42]. This conclusion is in line with the research results of this paper. There is a very complicated nonlinear relationship between the overall safety state of coal mining enterprises and various unsafe behaviors of miners [43]. Intervening unsafe behavior that can easily lead to coal mine accidents has a positive effect on the overall safety state of coal mines. The key work behaviors K1–K5 involved in H1 come from different groups of workers, including coal mine workers, field commanders, and management staff, all of which have a positive support effect on the prediction results. The results support Luria’s view that enhancing the visibility of workers’ behavior helps to increase the impact of intervention programs [44]. It is obvious that locating the key types of behavior intervention nodes is an effective way to enhance the visibility of workers’ behavior. Because of this remarkable influence relationship, in the daily coal mine safety management, in the prevention of roof accidents, we should pay attention to such unsafe behavior in the process of safety education and training, safety supervision, inspection, etc. [24], as this can effectively reduce the incidence of unsafe behavior.

The significant results of H2 support the positive effect of single-dimensional features on the intervention effect and verify the effectiveness of targeted intervention nodes S1–S7. Bona’s view shows that different interventions based on the distribution of risk levels of unsafe behavior among workers can achieve the best results at the lowest management cost [45]. Cheng studied the characteristics of occupational accidents in construction enterprises and found that more accidents occurred on the first day of workers’ presence in the workplace. According to the results of the time dimension analysis, enterprises can avoid accidents to a great extent [46]. In the study of human factors in coal mines in China, Chen found that environmental characteristics affected the occurrence of unsafe behavior to some extent. The environmental characteristics mentioned in this study are the spatial distribution of accidents determined by location, working conditions, geological characteristics, etc. It was also shown that mastering the position dimension distribution characteristics of unsafe behavior has a certain role in promoting the safety management efficiency of enterprises [47]. In the analysis of the unsafe behavior of the gas explosion accident in China, Yin et al. found that there are great differences in the frequency of the unsafe behavior of different types of work, which can make safety training easier and more effective [22]. This conclusion is consistent with the results of this paper. Sanmiquel et al. used a database of occupational accident and death reports in Spain’s mining industry to analyze the main causes of accidents. Some data mining techniques, such as Bayesian classifiers, decision trees, and contingency tables, were used to discover behavior patterns based on certain rules. The results were helpful in formulating appropriate preventive measures to reduce human injury and death [48]. In the study of road traffic accidents by Kumar, k-mode clustering technology and a correlation analysis algorithm were used to obtain combined characteristics of road traffic accidents. Through the trend analysis of road traffic accidents, it was found that the results of the study have a positive effect on reducing road traffic accidents [25]. The above conclusions are consistent with the results of this paper. The data distribution of dimension information such as risk level, time, position, etc. directly shows the single-dimensional characteristic of insecurity. According to the data distribution of each dimension of unsafe behavior, such dimensions can significantly change the overall safety climate of the enterprise by providing corresponding guidance to the intervention intensity and management resource allocation [49]. In the process of practical safety management of coal mine enterprises, the single-dimensional feature node should be taken as the basis for providing the guidance and basis for the intervention and resource allocation from the aspects of safety responsibility, safety culture, safety education, safety investment, and so on [50].

The positive results of H3 support the influence prediction between the multi-dimensional features and the intervention effect, which proves that the targeted intervention nodes M1–M6 can effectively improve the effect of behavioral intervention as the multi-dimensional feature nodes in the process of targeted intervention. The multi-dimensional intervention node determined by interaction analysis makes the targeted identification of the intervention objects more specific and the intervention process more precise [51]. The combinatorial localization of multi-dimensional information is a deep mining of unsafe behavior characteristics, which is different from general surface information and reveals the deep-seated characteristics of unsafe behavior [52]. 

The results of H4–H6 show that the three kinds of targeted intervention nodes also have interactive relationships, and the support of the analysis results for this prediction shows that the practical targeted intervention will be a dynamic process of intervention. When using single-dimensional features and multi-dimensional features to intervene, the key work behavior will change, and the single-dimensional feature information data distribution will also change when the intervention is aimed at multi-dimensional features. Therefore, in the process of targeted intervention in enterprises, intervention nodes are constantly changing. Because targeted intervention nodes are based on data analysis and positioning, this fully reflects the advantages of the data analysis method in the process of security management [53]. When implementing targeted intervention in enterprises, it is necessary to collect and update the unsafe behavior data of workers as the intervention continues, and to analyze the data regularly, so as to reposition the unsafe behavior targeted intervention nodes. Finally, in the process of enterprise safety management, the pertinence of worker unsafe behavior intervention is improved, and real-time intervention is realized.

## 5. Conclusions

The research of targeted interventions in other industries and different fields were combined with behavioral safety management and applied to unsafe behavior interventions. By dividing the data boundary, using safety management data in coal mine production and the method of single-dimensional statistical analysis and multi-dimensional interactive analysis, the targeted intervention process of miners’ unsafe behavior was constructed and deemed suitable for coal mines. In the study of miners’ unsafe behavior, this process realizes innovation from accident cases to data analysis and to behavior intervention. The method and process of the data analysis of behavior promote further development in the field of behavior intervention, and lay the theoretical foundation and method reference for the realization of accurate identification, accurate intervention, and accurate management of unsafe behavior.

The coal miners’ unsafe behavior was taken as the target intervention object. Through the analysis of the data of the coal mine roof accident, the characteristics of data distribution and the interaction rules in the accident case were excavated, and the target intervention nodes of the unsafe behavior were then located. The targeted intervention nodes can help to find unsafe behaviors that can easily lead to roof accidents, and provide a direction for enterprises to reduce the incidence of unsafe behaviors more quickly. According to the different proportion of intervention provided by intervention nodes, enterprises can optimize the allocation of management resources and improve management efficiency. Multiple intervention strategies provided by multiple intervention nodes have an effectively practical significance in controlling roof accidents in coal mines.

The evaluation model of targeted intervention nodes was established by using the structural equation method (SEM), based on which the evaluation process of unsafe behavior-targeted intervention nodes was formed. According to the evaluation results, the effectiveness of each target intervention node was determined, and the scientific and practical nature of the proposed method was further verified.

Based on the roof accident data, we studied the target intervention of miners’ unsafe behavior, and the research results have practical significance. The method and process can also be used to intervene on the unsafe behavior of workers in other industries. It is necessary to redefine the data boundary according to the behavior characteristics of workers in different industries, and make adaptive adjustments to the node location process by combining the sources of unsafe behavior data and the behavior characteristics of workers in various industries. This is a possible direction of future research and development.

## Figures and Tables

**Figure 1 ijerph-16-00422-f001:**
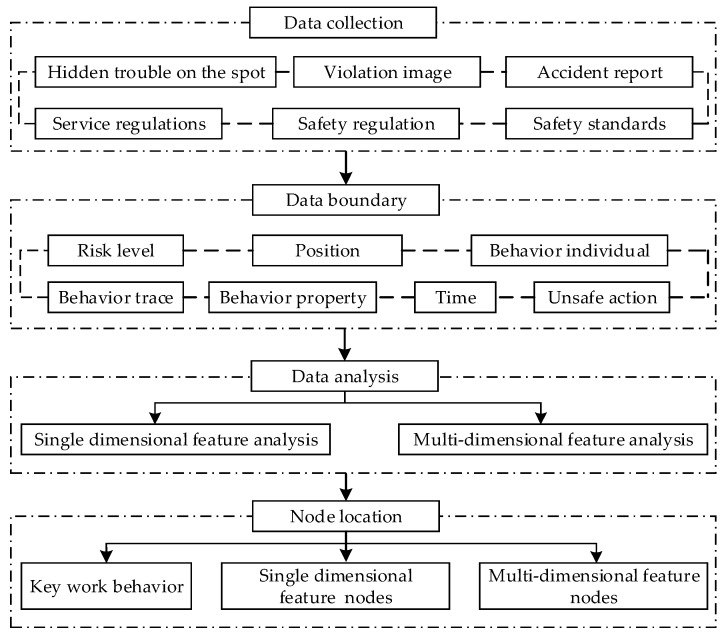
Process of targeted intervention node locating.

**Figure 2 ijerph-16-00422-f002:**
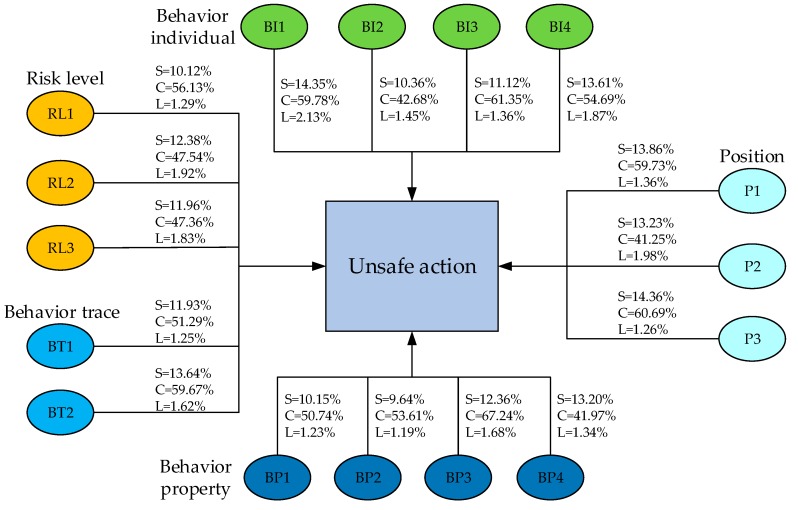
Association rule results of the miners’ unsafe behavior.

**Figure 3 ijerph-16-00422-f003:**
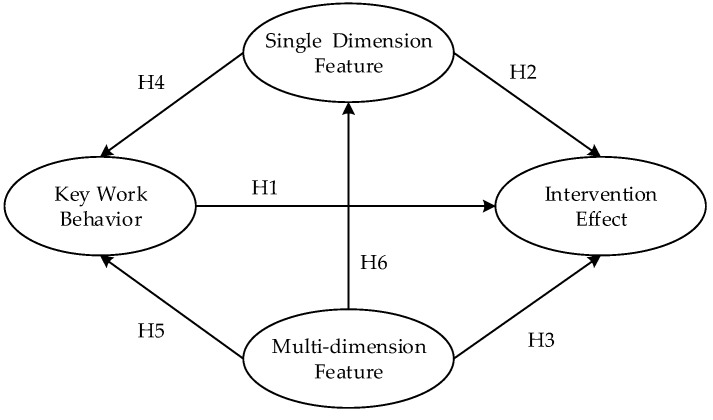
Initial model of miners’ targeted unsafe behavior intervention node evaluation.

**Figure 4 ijerph-16-00422-f004:**
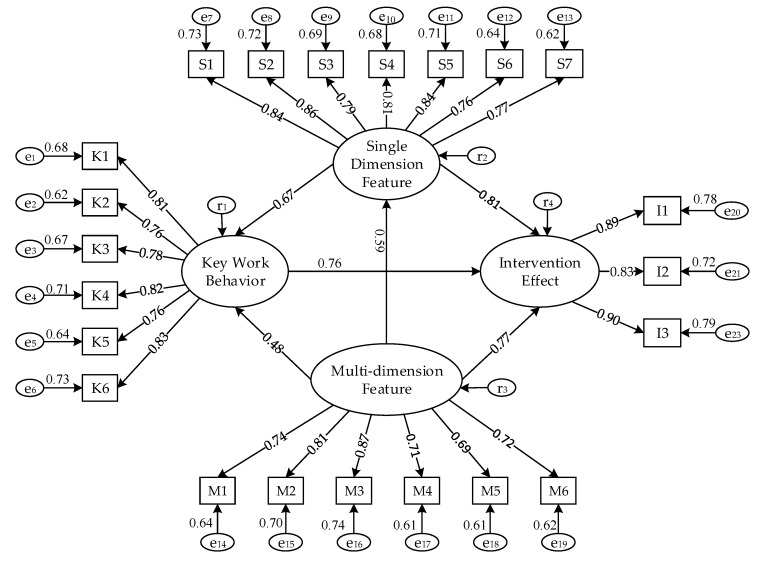
Final evaluation model of targeted intervention nodes for unsafe behavior.

**Table 1 ijerph-16-00422-t001:** Statistics of a single dimension.

Dimensional	Result (Frequency, Scale)
RL	High-risk (456, 37.5%); medium-risk (539, 44.4%); low-risk (220, 18.1%)
P	Coal face (504, 41.5%); tunneling working site (333, 27.4%); main roadway (198, 16.3%); others (180, 14.8%)
UA	Supporting (190, 18.36%); safety inspection (119, 11.50%); general type (114, 11.01%)
BI	Coal mining worker (628, 51.7%); field commanders (213, 17.5%); middle management staff (255, 21.0%); senior management staff (119, 9.8%)
BT	Traced behavior (574, 47.2%); non-traced behavior (641, 52.8%)
BP	Violation of action (706, 62.5%); violation of operation (316, 26.0%); violation of command (70, 1.4%); non-violation unsafe action (123, 10.1%)
T	More unsafe actions occur in January, March, and August.

Notes: RL: Risk Level; P: Position; UA: Unsafe action; BI: Behavior individual; BT: Behavior trace; BP: Behavior property; T: time.

**Table 2 ijerph-16-00422-t002:** Multidimensional interaction relationship analysis.

Dimension	Antecedent	Consequent	Association Rule
RL	High-risk	Empty roof operation	Different risk levels correspond to the most frequent miners’ unsafe action
	Medium-risk	No safety measures	
	Low-risk	Failure to clean up float coal in time	
P	Coal face	Inadequate supporting	Different workplaces correspond to the most frequent miners’ unsafe action
	Tunneling working site	Empty roof operation	
	Main roadway	Failure to strengthen roadway support	
BI	Coal mining worker	Empty roof operation	Different behavior individuals correspond to the most frequent miners’ unsafe action
	Field commanders	Inadequate supporting	
	Middle management staff	No site supervision of the workers’ work	
	Senior management staff	Illegal organization of production	
BT	Traced behavior	Inadequate supporting	The most frequent occurrence of the traced unsafe behavior is failure to support in time
	Non-traced behavior	The surrounding environment was not checked before the operation	The most frequent occurrence of the traced unsafe behavior is failure to check the surrounding environment before the operation
BP	Violation of action	Empty roof operation	Different behavior properties correspond to the most frequent miners’ unsafe action
	Violation of operation	Inadequate supporting	
	Violation of command	Illegal organization of production	
	Non-violation unsafe action	Nocking and drumming before the work are not careful	

**Table 3 ijerph-16-00422-t003:** List of miners’ unsafe behavior-targeted intervention nodes.

Types	Lable	Contents	
Key Work Behavior K	K1	Coal mining worker	Work under the empty roof
	K2	Safety inspection worker	No inspection of the working surface roof safety condition before the operation
	K3	Survey worker	The geological data around the coal mine were not updated in time
	K4	Field commander	No inspection of the work on the working field
	K5	Middle management staff	Safety technical measures were not made according to the situation of the working field before operation
	K6	Senior management staff	Illegal organization of production
Single-Dimensional Feature S	S1	Medium-risk > High-risk > Low-risk	Rational allocation of management resources according to the frequency of different risk behaviors
	S2	Coal face > Tunneling working site > main roadway > Others	Focus on observing coal face, tunneling working site, and roadway in daily behavior safety management
	S3	Supporting > Safety inspection > General type	Increasing the intervention of supporting and safety inspection work in daily behavior safety management
	S4	Coal mining worker > Middle management staff > Field commanders > Senior management staff	Rational allocation of safety training resources according to the frequency of different workers
	S5	Non-traced behavior > Traced behavior	More attention paid to the observation of non-traced behavior in daily safety management
	S6	Violation of action > Violation of operation > Non-violation unsafe action > Violation of command	Rational allocation of safety training and management resources according to the frequency of different workers
	S7	January, March, August	More attention paid in January, March, and August to safety management work
Multi-Dimensional Feature M	M1	High-risk → Empty roof operation	Emphasis placed on “controlling the empty roof operation” when intervening in high-risk behavior
	M2	Coal face → Inadequate supporting	Emphasis placed on “inadequate supporting” during the safety inspection at coal face
	M3	Main roadway→Failure to strengthen roadway support	Emphasis placed on “failure to strengthen roadway support” during the safety inspection at main roadway
	M4	Traced behavior → Inadequate supporting	Emphasis placed on “inadequate supporting” when intervening in traced behavior
	M5	Non-traced behavior →The Surrounding environment was not checked before the operation	Emphasis placed on “surrounding environment” “not checked before the operation” when intervening in non-traced behavior
	M6	Violation of command → Illegal organization of production	Emphasis placed on “illegal organization of production” when intervening the violation of command

**Table 4 ijerph-16-00422-t004:** Reliability test results for latent variables.

Latent Variable	Cronbach’s α	Cronbach’s α Based on Standardization Term	Number of Terms
Key work behavior node	0.813	0.837	6
Single-dimensional feature node	0.824	0.841	7
Multi-dimensional feature node	0.796	0.805	6
Intervention effect	0.811	0.828	3

**Table 5 ijerph-16-00422-t005:** Model fitting criteria and results.

Criteria	X^2^/df	RMSEA	NFI	RFI	IFI	TLI	CFI	PGFI
Fit Index	<3	<0.08	>0.9	>0.9	>0.9	>0.9	>0.9	>0.5
Model Index	3.062	0.068	0.923	0.906	0.912	0.956	0.914	0.587

Notes: RMSEA: Root-Mean-Square Error of Approximation; NFI: Comparative Fit Index; RFI: Relative Fit Index; IFI: Increasing Fit Index; TLI: Tucker-Lewis Index; CFI: Comparative Fit Index; PGFI: Goodness of Fit Index.

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
