# Peer review of "Evaluating Targeted Intervention on Coal Miners’ Unsafe Behavior"

_ijerph, 2019, doi:10.3390/ijerph16030422_

Round 1
Reviewer 1 Report
This is an interesting study, I have couple of concerns. Frist, the data collected is based on self statement from participants; however, to achieve the proposed objective authors need to collect data from actual behavior of miners. Using surveys is not a good practice to measure impact of interventions. Second, the behavior that needs to be changed is not very clear in the manuscript. Authors need to discuss it in more details.
Author Response
Response to Reviewer 1 Comments
Point 1: The data collected is based on self statement from participants; however, to achieve the proposed objective authors need to collect data from actual behavior of miners.

Response 1: Thank you for the opinion raised in the data collection, the concerns you mentioned are particularly important and constructive. In view of this problem, we have the following explanation:
First, the purpose of this paper is to eliminate the coal mine roof accidents, so a large amount of effective unsafe behavior data is required, the current identification technology of unsafe behavior cannot meet the number of data. Second, it is not completely certain whether unsafe behavior observed on site can be used as effective research samples. In the future work, we will pay attention to the identification of workers’ unsafe behavior, to make up for the existing defects.
In addition, based on your comments, we have improved the description of the source and collection of data. (Lines 215-219, Pages 6; Lines 231-232, Pages 6):
Previous coal mine accidents can provide enough data for the analysis of miners' unsafe behaviors, but the unsafe behaviors that lead to different types of accidents are different in nature. On the premise of considering to eliminate the heterogeneity of a large number of unsafe behavior data, this paper selects the unsafe behavior of miners leading to roof accidents of coal mine as the research object [25].
The data of miners' unsafe behavior is transformed from unstructured text record into structured data.
References:
25. Kumar, S.; Toshniwal, D. A data mining framework to analyze road accident data. Journal of Big Data 2015, 2(1):26, 10.1186/s40537-015-0035-y. Available online: https://doi.org/10.1186/s40537-015-0035-y. (accessed on 22 January 2019).
Point 2: Using surveys is not a good practice to measure impact of interventions.
Response 2: Thank you for your valuable advice. We further supported the results of this study by combining the research of others in Section 4. (Lines 385-387, Pages 13; Lines 393-396, Pages 13; Lines 401-423, Pages 13-14):
The results of Judi et al. show that precise positioning of specific unsafe behaviors and positive strengthening of specific safety practices can effectively reduce occupational accidents. This conclusion is in line with the research results of this paper [42].
The results support Luria's view that enhancing the visibility of workers' behavior Visibility helps to increase the impact of intervention programs [44]. It is obvious that locating the key types of behavior intervention nodes is an effective way to enhance the visibility of workers' behavior.
Bona's view shows that different interventions based on the distribution of risk levels of unsafe behavior among workers can achieve the best results at the lowest management cost [45]. Just as Cheng studied the characteristics of occupational accidents in construction enterprises, they found that more accidents occurred on the first day of the workers' presence in the workplace. According to the results of time dimension analysis, enterprises can avoid some accidents to a great extent [46]. In the study of human factors in coal mines in China, Chen found that the environmental characteristics affected the occurrence of unsafe behavior to some extent. The environmental characteristics mentioned in this study are the spatial distribution of accidents determined by location, working conditions, geological characteristics, etc. It also shows that mastering the position dimension distribution characteristics of unsafe behavior has a certain role in promoting the safety management efficiency of enterprises [47]. In the analysis of the unsafe behavior of the gas explosion accident in China, Yin et al. found that there is a great difference in the frequency of the unsafe behavior of different types of work, which can make the work of safety training easier and more effective [22]. This conclusion is consistent with the results of this paper. Sanmiquel et al. use a database of occupational accidents and deaths reports in Spain's mining industry to analysis the main causes of accidents. Some data mining techniques, such as Bayesian classifiers, decision trees and contingency tables, are used to discover behavior patterns based on certain rules. Finally, it is proved that the results are helpful in formulating appropriate preventive measures to reduce human injury and death [48]. In the study of road traffic accidents by Kumar, k-modes clustering technology and correlation analysis algorithm are used to obtain some combined characteristics of road traffic accidents. Through the trend analysis of road traffic accidents, it is found that, the results of the study have a positive effect on reducing road traffic accidents [25]. The above conclusions are consistent with the results of this paper.
References:
42. Komaki, J.; Barwick, K.; Scott, L. A behavioral approach to occupational safety: Pinpointing and reinforcing safe performance in a food manufacturing plant. Journal of Applied Psychology 1978, 63(4):434-445, 10.1037/0021-9010.63.4.434. Available online: https://doi.org/10.1037/0021-9010.63.4.434. (accessed on 22 January 2019).
44. Luria, G.; Zohar, D.; Erev, I. The effect of workers' visibility on effectiveness of intervention programs: Supervisory-based safety interventions. Journal of Safety Research 2008, 39(3):273-280, 10.1016/j.jsr.2007.12.003. Available online: https://doi.org/10.1016/j.jsr.2007.12.003. (accessed on 22 January 2019).
45. Di Bona, G.; Silvestri, A.; Forcina, A. et al. Total efficient risk priority number (TERPN): a new method for risk assessment. Journal of Risk Research 2017, 1-25, 10.1080/13669877.2017.1307260. Available online: https://doi.org/10.1080/13669877.2017.1307260. (accessed on 22 January 2019).
46. Cheng, C.; Leu, S.; Lin, C. et al. Characteristic analysis of occupational accidents at small construction enterprises. Safety Science 2010, 48(6):698-707, 10.1016/j.ssci.2010.02.001. Available online: https://doi.org/10.1016/j.ssci.2010.02.001. (accessed on 22 January 2019).
47. Chen, H.; Qi, H.; Long, R. et al. Research on 10-year tendency of China coal mine accidents and the characteristics of human factors. Safety Science 2012, 50(4):745-750, 10.1016/j.ssci.2011.08.040. Available online: https://doi.org/10.1016/j.ssci.2011.08.040. (accessed on 22 January 2019).
22. Yin, W.; Fu, G.; Yang, C.; et al. Fatal gas explosion accidents on Chinese coal mines and the characteristics of unsafe behaviors: 2000–2014. Safety Science 2017, 92(Complete), 173-179, 10.1016/j.ssci.2016.09.018. Available online: https://doi.org/10.1016/j.ssci.2016.09.018. (accessed on 2 December 2018).
48. Sanmiquel, LluĂs.; Rossell, J.; VintrĂł, Carla. Study of Spanish mining accidents using data mining techniques. Safety Science 2015, 75:49-55, 10.1016/j.ssci.2015.01.016. Available online: https://doi.org/10.1016/j.ssci.2015.01.016. (accessed on 22 January 2019).
25. Kumar, S.; Toshniwal, D. A data mining framework to analyze road accident data. Journal of Big Data 2015, 2(1):26, 10.1186/s40537-015-0035-y. Available online: https://doi.org/10.1186/s40537-015-0035-y. (accessed on 22 January 2019).
Point 3: The behavior that needs to be changed is not very clear in the manuscript. Authors need to discuss it in more details.
Response 2: Thank you for your comments on the description of unsafe behavior in this study. We added the descriptions of unsafe behavior that needs to be changed. (Lines 205-217, Pages 6; Lines 221-229, Pages 6):
Unsafe behavior refers to the behavior of the person who has caused the accident or may cause the accident, which is the direct cause of the accident. There are many reasons leading to unsafe behaviors, including individual factors, psychological factors, organizational factors, environmental factors and so on. Different researchers have different views on the classification of unsafe behaviors. Unsafe behaviors in the field of coal mine mainly refer to the "three disobeying" of coal mine safety production [24]. "Three disobeying" is the general term of disobeying command, disobeying operation and disobeying labor discipline. The elimination of "three disobeying" has always been an important issue in the safety management of all industries, especially coal mining enterprises and other high-risk industries. In order to fundamentally explore the objective rule of unsafe behaviors and the complex relationship between its internal factors, a large number of unsafe behaviors need to be objectively analyzed. Previous coal mine accidents can provide enough data for the analysis of miners' unsafe behaviors, but the unsafe behaviors that lead to different types of accidents are different in nature. On the premise of considering to eliminate the heterogeneity of a large number of unsafe behavior data, this paper selects the unsafe behavior of miners leading to roof accidents of coal mine as the research object [25].
All unsafe behaviors of miners causing roof accidents are collected from the accident report. For example, a roof accident occurred in a mine in Pingdingshan city, Henan province which directly resulted in one death and several serious injuries, resulting in a total economic lost about RMB 585,000. According to the time of this accident, four unsafe behaviors can be analyzed successively, namely, the “The top plate is out of the slag, and it is not withdrawn in time.” related to the coal miner, the “No timely measures to prevent potential safety hazards” related to the field commanders, the “No perfect operation procedures and safety technical measures have been established” related to the middle management staff, and the “arrangement of roadway in a steep-inclined coal seam” related to the senior management staff.
References:
24. Cao, Q.; Li, K.; Liu, Y.; et al. Risk Management and Workers' Safety Behavior Control in Coal Mines. International Symposium on Mine Safety Science & Engineering 2012, 50, 909-913, 10.1016/j.ssci.2011.08.005. Available online: https://doi.org/10.1016/j.ssci.2011.08.005. (accessed on 2 December 2018).
25. Kumar, S.; Toshniwal, D. A data mining framework to analyze road accident data. Journal of Big Data 2015, 2(1):26, 10.1186/s40537-015-0035-y. Available online: https://doi.org/10.1186/s40537-015-0035-y. (accessed on 22 January 2019).

Reviewer 2 Report
The new model is builded on an appropriate base of theory. The results are presented clearly and analysed appropriately. The authors have to better explained the implication for research. The language used by the authors in writing the paper is satisfactory when considering the grammar, spelling and punctuation marks. The referecens have to be improved, e.g.:
- An Analytical Model to Measure the Effectiveness of Safety Management Systems: Global Safety Improve Risk Assessment (G-SIRA) Method
G. Di Bona, A.Silvestri, A. Forcina, A. Petrillo, F. De FeliceJournal of Failure Analysis and Prevention ISSN 1547-7029 DOI 10.1007/s11668-016-0185-z Volume 16, Issue 6, December 2016 Pages 1024-1037
- Total Efficient Risk Priority Number (TERPN): a new method for risk assessment G. Di Bona, A. Forcina, A Silvestri, A.PetrilloJournal of Risk Research 2017 ISSN: 1366-9877 http://dx.doi.org/10.1080/13669877.2017.1307260 (article online)
Author Response
Response to Reviewer 2 Comments
Point 1: The new model is builded on an appropriate base of theory. The results are presented clearly and analysed appropriately. The authors have to better explained the implication for research. The language used by the authors in writing the paper is satisfactory when considering the grammar, spelling and punctuation marks. The references have to be improved.

Response 1: Thank you for your opinion on the reference. We have added 12 new references to the article, for example (Lines 43-45, Pages 2; Lines 401-403, Pages 13):
Reducing the incidence of unsafe behavior among workers through behavior intervention is conducive to reducing the risk of enterprise safety, thus enhancing the effectiveness of the enterprise safety management system [8].
Bona's view shows that different interventions based on the distribution of risk levels of unsafe behavior among workers can achieve the best results at the lowest management cost [45].
References:
8. Di Bona, G.; Silvestri A.; Forcina, A. et al. An Analytical Model to Measure the Effectiveness of Safety Management Systems: Global Safety Improve Risk Assessment (G-SIRA) Method. Journal of Failure Analysis & Prevention 2016, 16(6):1-14, 10.1007/s11668-016-0185-z. Available online: https://doi.org/10.1007/s11668-016-0185-z. (accessed on 22 January 2019).
45. Di Bona, G.; Silvestri, A.; Forcina, A. et al. Total efficient risk priority number (TERPN): a new method for risk assessment. Journal of Risk Research 2017, 1-25, 10.1080/13669877.2017.1307260. Available online: https://doi.org/10.1080/13669877.2017.1307260. (accessed on 22 January 2019).

Reviewer 3 Report
At first, this paper seems to be a report, but not a scientific paper to me, since there are lots of fundamental introduction in the paper. For example, introduction on SEM, and introduction for targeted intervention nodes evaluation. Such information makes the audience easily lost their focus. The research foundations part might be removed from the paper.
There are several statements in the manuscript without strong support. For example, in the abstract, "lack of pertinence" is rather vague. There is no specific scientific problem clearly written in the manuscript.
The authors have a special focus in Mining industry, however, the authors failed to present why conventional method are not suitable for solving the safety management problem, and how the approach proposed by the authors could be distinguished from conventional method. The strength of their new approach should be supported with evidence, but not only a structural equation model.
The introduction and the whole paper has no flow, which makes it difficult to follow. I strongly recommend that the authors should clarify their logic and their major contribution, so that the readers could understand the points from the authors.
Author Response
Response to Reviewer 3 Comments
Point 1: At first, this paper seems to be a report, but not a scientific paper to me, since there are lots of fundamental introduction in the paper. For example, introduction on SEM, and introduction for targeted intervention nodes evaluation. Such information makes the audience easily lost their focus. The research foundations part might be removed from the paper.

Response 1: Thank you for your suggestions about the content of the article. We accepted it, removed some fundamental introduction in the paper, and made a little adjustment to the structure of the article. (Lines 169-194, Pages 5-6; Lines 281-300, Pages 10):
Point 2: There are several statements in the manuscript without strong support. For example, in the abstract, "lack of pertinence" is rather vague. There is no specific scientific problem clearly written in the manuscript.
Response 2: Thank you for your valuable advice. We reinforce the description of "lack of pertinence" and add references to support it in Section introduction. (Lines 70-77, Pages 2):
These traditional methods of unsafe behavior intervention are mostly based on the observation and record of workers' behaviors or the assessment results of behavioral risks to conduct overall and extensive behavior intervention for workers. In general, traditional methods are only for the intervention of unsafe behavior itself, lacking in-depth analysis of its internal characteristics and in-depth study of the root factors leading to unsafe behavior [16]. Therefore, this intervention method lacks pertinence. People's unsafe behavior can’t be fundamentally changed, and the root factors of unsafe behavior will reappear after the intervention [17].
References:
16. Johnson, S.; Hall, A. The prediction of safe lifting behavior: An application of the theory of planned behavior. Journal of Safety Research 2005, 36(1):63-73, 10.1016/j.jsr.2004.12.004. Available online: https://doi.org/10.1016/j.jsr.2004.12.004. (accessed on 22 January 2019).
17. Lee, Y.; Kim, Y.; Kim, S. H.; et al. Analysis of human error and organizational deficiency in events considering risk significance. Nuclear Engineering & Design 2004, 230(1):61-67, 10.1016/j.nucengdes.2003.11.019. Available online: https://doi.org/10.1016/j.nucengdes.2003.11.019. (accessed on 22 January 2019).
Point 3: The authors have a special focus in Mining industry, however, the authors failed to present why conventional method are not suitable for solving the safety management problem, and how the approach proposed by the authors could be distinguished from conventional method.
Response 3: Thank you for this suggestion, which can strengthen the research base of our article. We added the shortcomings of the traditional method and the direction that needed to be improved. (Lines 59-72, Pages 2):
Namian et al. found that prolonged use of ineffective safety training methods seriously affected safety in the building. They collected safety training data at the project level to measure workers' risk identification ability and safety risk perception level, and the results provided references for improving safety training work. It shows that traditional safety training methods need to be combined with modern information technology to enhance their effectiveness [13]. Kouabenan's research on two French nuclear power plants proves that the safety climate has a certain substantial impact on the promotion of safety management, but also finds that the encouragement of the direct supervisor in the enterprise is more influential than the view of senior management on safety. This also indicates that more in-depth research on human behavior characteristics is needed in enterprise security management [14]. Warszawska et al. found that weak safety culture is the main cause of many catastrophic events, and in order to avoid this situation, the construction of enterprise safety culture must be strengthened [15]. These traditional methods of unsafe behavior intervention are mostly based on the observation and record of workers' behaviors or the assessment results of behavioral risks to conduct overall and extensive behavior intervention for workers.
References:
13. Mostafa, N.; Alex, A.; Carlos, M.; et al. Role of Safety Training: Impact on Hazard Recognition and Safety Risk Perception. Journal of construction engineering and management 2016, 142(12), 164, 10.1061/(ASCE)CO.1943-7862.0001198. Available online: https://doi.org/10.1061/(ASCE)CO.1943-7862.0001198. (accessed on 22 January 2019).
14. Kouabenan, D.; Ngueutsa, R.; Mbaye, S. Safety climate, perceived risk, and involvement in safety management. Safety Science 2015, 77:72-79, 10.1016/j.ssci.2015.03.009. Available online: https://doi.org/10.1016/j.ssci.2015.03.009. (accessed on 22 January 2019).
15. Warszawska, K.; Kraslawski, A. Method for quantitative assessment of safety culture. Journal of Loss Prevention in the Process Industries 2016, 42:27-34, 10.1016/j.jlp.2015.09.005. Available online: https://doi.org/10.1016/j.jlp.2015.09.005. (accessed on 22 January 2019).
Point 4: The strength of their new approach should be supported with evidence, but not only a structural equation model.
Response 4: Thank you for this suggestion, which can enhance the persuasiveness of our article. We further supported the results of this study by combining the research of others in Section 4. (Lines 385-387, Pages 13; Lines 393-395, Pages 13; Lines 401-423, Pages 13-14):
The results of Judi et al. show that precise positioning of specific unsafe behaviors and positive strengthening of specific safety practices can effectively reduce occupational accidents. This conclusion is in line with the research results of this paper [42].
The results support Luria's view that enhancing the visibility of workers' behavior helps to increase the impact of intervention programs [44]. It is obvious that locating the key types of behavior intervention nodes is an effective way to enhance the visibility of workers' behavior.
Bona's view shows that different interventions based on the distribution of risk levels of unsafe behavior among workers can achieve the best results at the lowest management cost [45]. Just as Cheng studied the characteristics of occupational accidents in construction enterprises, they found that more accidents occurred on the first day of the workers' presence in the workplace. According to the results of time dimension analysis, enterprises can avoid some accidents to a great extent [46]. In the study of human factors in coal mines in China, Chen found that the environmental characteristics affected the occurrence of unsafe behavior to some extent. The environmental characteristics mentioned in this study are the spatial distribution of accidents determined by location, working conditions, geological characteristics, etc. It also shows that mastering the position dimension distribution characteristics of unsafe behavior has a certain role in promoting the safety management efficiency of enterprises [47]. In the analysis of the unsafe behavior of the gas explosion accident in China, Yin et al. found that there is a great difference in the frequency of the unsafe behavior of different types of work, which can make the work of safety training easier and more effective [22]. This conclusion is consistent with the results of this paper. Sanmiquel et al. use a database of occupational accidents and deaths reports in Spain's mining industry to analysis the main causes of accidents. Some data mining techniques, such as Bayesian classifiers, decision trees and contingency tables, are used to discover behavior patterns based on certain rules. Finally, it is proved that the results are helpful in formulating appropriate preventive measures to reduce human injury and death [48]. In the study of road traffic accidents by Kumar, k-modes clustering technology and correlation analysis algorithm are used to obtain some combined characteristics of road traffic accidents. Through the trend analysis of road traffic accidents, it is found that, the results of the study have a positive effect on reducing road traffic accidents [25]. The above conclusions are consistent with the results of this paper.
References:
42. Komaki, J.; Barwick, K.; Scott, L. A behavioral approach to occupational safety: Pinpointing and reinforcing safe performance in a food manufacturing plant. Journal of Applied Psychology 1978, 63(4):434-445, 10.1037/0021-9010.63.4.434. Available online: https://doi.org/10.1037/0021-9010.63.4.434. (accessed on 22 January 2019).
44. Luria, G.; Zohar, D.; Erev, I. The effect of workers' visibility on effectiveness of intervention programs: Supervisory-based safety interventions. Journal of Safety Research 2008, 39(3):273-280, 10.1016/j.jsr.2007.12.003. Available online: https://doi.org/10.1016/j.jsr.2007.12.003. (accessed on 22 January 2019).
45. Di Bona, G.; Silvestri, A.; Forcina, A. et al. Total efficient risk priority number (TERPN): a new method for risk assessment. Journal of Risk Research 2017, 1-25, 10.1080/13669877.2017.1307260. Available online: https://doi.org/10.1080/13669877.2017.1307260. (accessed on 22 January 2019).
46. Cheng, C.; Leu, S.; Lin, C. et al. Characteristic analysis of occupational accidents at small construction enterprises. Safety Science 2010, 48(6):698-707, 10.1016/j.ssci.2010.02.001. Available online: https://doi.org/10.1016/j.ssci.2010.02.001. (accessed on 22 January 2019).
47. Chen, H.; Qi, H.; Long, R. et al. Research on 10-year tendency of China coal mine accidents and the characteristics of human factors. Safety Science 2012, 50(4):745-750, 10.1016/j.ssci.2011.08.040. Available online: https://doi.org/10.1016/j.ssci.2011.08.040. (accessed on 22 January 2019).
22. Yin, W.; Fu, G.; Yang, C.; et al. Fatal gas explosion accidents on Chinese coal mines and the characteristics of unsafe behaviors: 2000–2014. Safety Science 2017, 92(Complete), 173-179, 10.1016/j.ssci.2016.09.018. Available online: https://doi.org/10.1016/j.ssci.2016.09.018. (accessed on 2 December 2018).
48. Sanmiquel, LluĂs.; Rossell, J.; VintrĂł, Carla. Study of Spanish mining accidents using data mining techniques. Safety Science 2015, 75:49-55, 10.1016/j.ssci.2015.01.016. Available online: https://doi.org/10.1016/j.ssci.2015.01.016. (accessed on 22 January 2019).
48. Kumar, S.; Toshniwal, D. A data mining framework to analyze road accident data. Journal of Big Data 2015, 2(1):26, 10.1186/s40537-015-0035-y. Available online: https://doi.org/10.1186/s40537-015-0035-y. (accessed on 22 January 2019).
Point 5: The introduction and the whole paper has no flow, which makes it difficult to follow. I strongly recommend that the authors should clarify their logic and their major contribution, so that the readers could understand the points from the authors.
Response 5: Thank you for this suggestion, which can enhance the coherence and logic of the article. We have adjusted and supplemented the introduction. The first paragraph proposed to strengthen the research on behavior intervention according to the current situation of roof accident prevention. The second paragraph points out the shortcomings and the direction to be improved of traditional behavior intervention methods. The third paragraph proposed a new behavior intervention method and the main research contents of this paper. (Lines 26-103, Pages 13):
At present, roof accidents are still a kind of frequent accidents in coal mine production. According to the previous analysis of roof accidents in coal mines, it is found that the engineering technical means can’t completely control the occurrence of roof accidents. However, the research on the influence of human behavior is not clear [1-3]. Currently, the research on roof accidents is mainly divided into two categories: management and technology. The former does not point out the concrete operation mistake or the management mistake, has caused the barrier to the formulation pertinence measure. The latter combined with specific coal mining face or roadway, therefore the mechanism of roof deformation and failure instability is studied, and relevant engineering measures are worked out [4]. However, despite the continuous improvement of the technical level of roof support in China, the roof accidents still occur from time to time. This shows that engineering technical means are not the fundamental measures to solve roof accidents. With the continuous improvement and deepening of people's understanding and research on behavior safety, researchers have gradually discovered that unsafe behavior is the more important cause of accidents [5]. Similarly, the vast majority of coal mine casualties are caused by unsafe behavior by miners [6]. Unsafe behavior refers to those who may cause casualties, property damage, environmental damage in violation of the rules of operation, safety regulations. In order to effectively prevent roof accidents, the human factors must be taken into account. Combined with BBS theory, explicit behavioral intervention measures should be put forward [7]. Reducing the incidence of unsafe behavior among workers through behavior intervention is conducive to reducing the risk of enterprise safety, thus enhancing the effectiveness of the enterprise safety management system [8].
Miner’s behavior is a complex decision-making process, and behavior safety research is mainly based on objective theory and operational condition theory, the emphasis is to identify key unsafe behavior and correct unsafe behavior through intervention. Currently, the research on unsafe behavior intervention mainly explores the intervention countermeasures of unsafe behavior by combining organizational factors with individual factors [9]. According to the planning behavior theory, accident cause theory, structural equation model and other methods, the researchers in coal mine, building, aviation and other fields, respectively, from safety training, safety culture, performance feedback, material incentives and other aspects of appropriate intervention measures to reduce the occurrence of unsafe behavior [10-12]. Namian et al. found that prolonged use of ineffective safety training methods seriously affected safety in the building. They collected safety training data at the project level to measure workers' risk identification ability and safety risk perception level, and the results provided references for improving safety training work. It shows that traditional safety training methods need to be combined with modern information technology to enhance their effectiveness [13]. Kouabenan's research on two French nuclear power plants proves that the safety climate has a certain substantial impact on the promotion of safety management, but also finds that the encouragement of the direct supervisor in the enterprise is more influential than the view of senior management on safety. This also indicates that more in-depth research on human behavior characteristics is needed in enterprise security management [14]. Warszawska et al. found that weak safety culture is the main cause of many catastrophic events, and in order to avoid this situation, the construction of enterprise safety culture must be strengthened [15]. These traditional methods of unsafe behavior intervention are mostly based on the observation and record of workers' behaviors or the assessment results of behavioral risks to conduct overall and extensive behavior intervention for workers. In general, traditional methods are only for the intervention of unsafe behavior itself, lacking in-depth analysis of its internal characteristics and in-depth study of the root factors leading to unsafe behavior [16]. Therefore, this intervention method lacks pertinence. People's unsafe behavior can’t be fundamentally changed, and the root factors of unsafe behavior will reappear after the intervention [17].
With the continuous expansion of the data volume of security information, analyzing and mining the hidden value of the data has become an important field of behavior security research [18]. The purpose of this study is to propose a targeted intervention method for unsafe behavior on the basis of behavior safety management. This method pays more attention to the inherent characteristics of workers' unsafe behaviors on the basis of traditional intervention methods, through the data mining to fully master the risk level, position, behavior individual, behavior trace, behavioral property, time and type of unsafe action dimension information for worker’s unsafe behavior, statistical analysis and association rule mining method is applied to analyze various dimension information of distribution and the inner link between them. According to the internal information of unsafe behaviors, the intervention nodes are located. The ultimate goal is to improve the effect of behavioral intervention by formulating corresponding intervention measures for each targeted intervention node in the actual safety management work. Through the data analysis of coal mine roof accident cases, three types of targeted intervention nodes are identified: the key types of behavior targeted intervention nodes, the single dimension characteristic targeted intervention nodes, and the multi-dimension characteristic targeted intervention nodes. The structural equation model (SEM) was used to evaluate each targeted intervention node to prove the effectiveness and practicability of the method.
References:
1. Jiang, W.; Qu, F.; Zhang, L. Quantitative identification and analysis on hazard sources of roof fall accident in coal mine. Procedia Engineering 2012, 45(3), 83-88, 10.1016/j.proeng.2012.08.125. Available online: https://doi.org/10.1016/j.proeng.2012.08.125. (accessed on 2 December 2018).
2. H.S.B, D. Analysis of roof fall hazards and risk assessment for zonguldak coal basin underground mines, International Journal of Coal Geology, 2005, 64(1-2), 104-115, 10.1016/j.coal.2005.03.008. Available online: https://doi.org/10.1016/j.coal.2005.03.008. (accessed on 2 December 2018).
3. Yang, D.; Zhang, L.; Chai, M.; et al. Study of roof breaking law of fully mechanized top coal caving mining in ultra-thick coal seam based on fracture mechanics. Rock & Soil Mechanics 2016, 37(7), 2033-2039, 10.16285/j.rsm.2016.07.026. Available online: https://doi.org/10.16285/j.rsm.2016.07.026. (accessed on 2 December 2018).
4. Wang, W.; Luo, L. Yu, W. Study of dynamic pressure roadway supporting scheme under condition of thick composite roof. Journal of Coal Science and Engineering (China) 2013, 19(2), 119-125, 10.1007/s12404-013-0202-8. Available online: https://doi.org/10.1007/s12404-013-0202-8. (accessed on 2 December 2018).
5. Cooper, M. Exploratory Analyses of the Effects of Managerial Support and Feedback Consequences on Behavioral Safety Maintenance. Journal of Organizational Behavior Management 2006, 26(3), 1-41, 10.1300/j075v26n03_01. Available online: https://doi.org/10.1300/j075v26n03_01. (accessed on 2 December 2018).
6. Paul, P.; Maiti, J. The role of behavioral factors on safety management in underground mines. Safety Science 2007, 45(4), 449-471, 10.1016/j.ssci.2006.07.006. Available online: https://doi.org/10.1016/j.ssci.2006.07.006. (accessed on 2 December 2018).
7. Chen, D.; Tian, H. Behavior based safety for accident prevention and positive study in China construction projects. Procedia Engineering 2012, 43, 528-534, 10.1016/j.proeng.2012.08.092. Available online: https://doi.org/10.1016/j.proeng.2012.08.092. (accessed on 2 December 2018).
8. Di Bona, G.; Silvestri A.; Forcina, A. et al. An Analytical Model to Measure the Effectiveness of Safety Management Systems: Global Safety Improve Risk Assessment (G-SIRA) Method. Journal of Failure Analysis & Prevention 2016, 16(6):1-14, 10.1007/s11668-016-0185-z. Available online: https://doi.org/10.1007/s11668-016-0185-z. (accessed on 22 January 2019).
9. Priebe, R.; Thomasolson, L. An exploration and analysis on the timeliness of critical incident stress management interventions in healthcare. International Journal of Emergency Mental Health 2013, 15(1), 39-49,
10.1016/j.tree.2010.07.010. Available online: https://doi.org/10.1016/j.tree.2010.07.010. (accessed on 2 December 2018).
10. Fogarty, G.; Shaw, A. Safety Climate and the Theory of Planned Behavior: Towards the Prediction of Unsafe Behavior. Accident Analysis and prevention 2010, 42(5), 1455-1459, 10.1016/j.aap.2009.08.008. Available online: https://doi.org/10.1016/j.aap.2009.08.008. (accessed on 2 December 2018).
11. Clarke, S. Contrasting perceptual, attitudinal and dispositional approaches to accident involvement in the workplace. Safety Science 2006, 44(6), 537-550, 10.1016/j.ssci.2005.12.001. Available online: https://doi.org/10.1016/j.ssci.2005.12.001. (accessed on 2 December 2018).
12. Aryee, S.; Hsiung, H. Regulatory focus and safety outcomes: an examination of the mediating influence of safety behavior. Safety Science 2016, 86, 27-35, 10.1016/j.ssci.2016.02.011. Available online: https://doi.org/10.1016/j.ssci.2016.02.011. (accessed on 2 December 2018).
13. Mostafa, N.; Alex, A.; Carlos, M.; et al. Role of Safety Training: Impact on Hazard Recognition and Safety Risk Perception. Journal of construction engineering and management 2016, 142(12), 04016073, 10.1061/(ASCE)CO.1943-7862.0001198. Available online: https://doi.org/10.1061/(ASCE)CO.1943-7862.0001198. (accessed on 22 January 2019).
14. Kouabenan, D.; Ngueutsa, R.; Mbaye, S. Safety climate, perceived risk, and involvement in safety management. Safety Science 2015, 77:72-79, 10.1016/j.ssci.2015.03.009. Available online: https://doi.org/10.1016/j.ssci.2015.03.009. (accessed on 22 January 2019).
15. Warszawska, K.; Kraslawski, A. Method for quantitative assessment of safety culture. Journal of Loss Prevention in the Process Industries 2016, 42:27-34, 10.1016/j.jlp.2015.09.005. Available online: https://doi.org/10.1016/j.jlp.2015.09.005. (accessed on 22 January 2019).
16. Johnson, S.; Hall, A. The prediction of safe lifting behavior: An application of the theory of planned behavior. Journal of Safety Research 2005, 36(1):63-73, 10.1016/j.jsr.2004.12.004. Available online: https://doi.org/10.1016/j.jsr.2004.12.004. (accessed on 22 January 2019).
17. Lee, Y.; Kim, Y.; Kim, S. H.; et al. Analysis of human error and organizational deficiency in events considering risk significance. Nuclear Engineering & Design 2004, 230(1):61-67, 10.1016/j.nucengdes.2003.11.019. Available online: https://doi.org/10.1016/j.nucengdes.2003.11.019. (accessed on 22 January 2019).
18. Geller, E. Behavior-based safety and occupational risk management. Behavior Modification 2005, 29(3), 539, 10.1177/0145445504273287. Available online: https://doi.org/10.1177/0145445504273287. (accessed on 2 December 2018).

Round 2
Reviewer 1 Report
No further comment.